# Evaluation of Two *EGFR* Mutation Tests on Tumor and Plasma from Patients with Non-Small Cell Lung Cancer

**DOI:** 10.3390/cancers12040785

**Published:** 2020-03-26

**Authors:** Jeong-Oh Kim, Jung-Young Shin, Seo Ree Kim, Kab Soo Shin, Joori Kim, Min-Young Kim, Mi-Ran Lee, Yonggoo Kim, Myungshin Kim, Sook Hee Hong, Jin Hyoung Kang

**Affiliations:** 1Laboratory of Medical Oncology, Cancer Research Institute, College of Medicine, The Catholic University of Korea, Seoul 06591, Korea; kjo9713@catholic.ac.kr (J.-O.K.); bearjy@catholic.ac.kr (J.-Y.S.); kiminy414@catholic.ac.kr (M.-Y.K.); miran13@catholic.ac.kr (M.-R.L.); 2Department of Medical Oncology, Seoul St. Mary’s Hospital, The Catholic University of Korea, Seoul 06591, Korea; seoreek@gmail.com (S.R.K.); agx002@naver.com (K.S.S.); jooriworld@gmail.com (J.K.); ssuki76@catholic.ac.kr (S.H.H.); 3Department of Laboratory Medicine, Seoul St. Mary’s Hospital, The Catholic University of Korea, Seoul 06591, Korea; yonggoo@catholic.ac.kr (Y.K.); microkim@catholic.ac.kr (M.K.)

**Keywords:** circulating free DNA, liquid biopsy, epidermal growth factor receptor, tyrosine kinase inhibitor, osimertinib

## Abstract

Epidermal growth factor receptor (*EGFR*) mutation testing is essential for individualized treatment using tyrosine kinase inhibitors. We evaluated two *EGFR* mutation tests, cobas v2 and PANAMutyper, for detection of *EGFR* activating mutations Ex19del, L858R, and T790M in tumor tissue and plasma from 244 non-small cell lung cancer (NSCLC) patients. The Kappa coefficient (95% CI) between the tests was 0.82 (0.74–0.92) in tumor samples (suggesting almost perfect agreement) and 0.69 (0.54–0.84) in plasma (suggesting substantial agreement). In plasma samples, both tests showed low to moderate sensitivity depending on disease stage but high diagnostic precision (86%–100%) in all disease stages (sensitivity: percentage of mutations in tumors that are also detected in plasma; precision: percentage of mutations in plasma which are also detected in tumors). Among the 244 patients, those previously diagnosed as T790M carriers who received osimertinib treatment showed dramatically better clinical outcomes than T790M carriers without osimertinib treatment. Taken together, our study supports interchangeable use of cobas v2 and PANAMutyper in tumor and plasma *EGFR* testing. Both tests have high diagnostic precision in plasma but are particularly valuable in late-stage disease. Our clinical data in T790M carriers strongly support the clinical benefits of osimertinib treatment guided by both *EGFR* mutation tests.

## 1. Introduction

The introduction of tyrosine kinase inhibitors (TKIs) for non-small cell lung cancer (NSCLC) has greatly improved treatment outcomes in patients with epidermal growth factor receptor (*EGFR*) mutations [1,2,3]. The importance of *EGFR* mutation testing in TKI treatment is well-recognized [4], and its cost-effectiveness has been established in many countries [5,6,7]. Inaccurate *EGFR* mutation tests may cause marked loss of quality-adjusted life-years [8]. Direct sequencing of tumor DNA is the gold standard diagnostic method for detecting *EGFR* mutations, but clinical utility is limited due to its high cost, long turnaround time, and low sensitivity (limit of detection >20%; limit of detection is defined as the percentage of tumor cells that must be present in the specimen for a mutation to be identified [4]). New *EGFR* mutation tests such as real-time polymerase chain reaction (RT-PCR), amplification refractory mutation system (ARMS), peptide nucleic acid (PNA)-based PCR, and droplet digital PCR (dd-PCR) provide reliable, quick test results with high sensitivity (limit of detection: 0.01%–1%) and have therefore gained currency in clinical settings [9]. Meanwhile, due to the difficulty in obtaining sufficient amounts of tumor tissue and repeat tumor biopsy, there is a growing trend of testing *EGFR* mutations using liquid biopsy.

The objective of this study was to evaluate the concordance of two commercial *EGFR* mutation tests: an RT-PCR method cobas EGFR mutation test v2 (cobas v2) and a PNA-based PCR method PANAMutyper R EGFR (PANAMutyper) in tumor tissue and plasma from NSCLC patients. The cobas v2 was initially approved by the US Food Drug Administration, and the PANAMutyper was initially approved by the Korea Ministry of Food and Drug Safety. The PANAMutyper test combines PNA-based PCR clamping (PNAClamp) with multiplex fluorescence melting curve analysis (PANA S-Melting) using a fluorescence-labeled PNA probe, which allows detection of 47 hotspot mutations between *EGFR* exon 18 and exon 21 [10]. While evaluating the concordance of the tests, we particularly focused on the performance of the tests in plasma samples and in a subgroup of patients with diagnosis of *EGFR* T790M mutation (An acquired TKI resistance-related *EGFR* gatekeeper mutation, which substitutes a threonine with a methionine at position 790 of exon 20).

## 2. Materials and Methods

### 2.1. Study Population

Electronic medical records (EMRs) were used to identify eligible patients histologically diagnosed with NSCLC between January 2013 and April 2019 with tumor tissue or plasma sample stored at the Seoul St. Mary’s Hospital Biobank. When multiple samples were available from the same patient, samples from the first biopsy were used for testing. Almost all of the eligible patients had been tested for *EGFR* mutations using PNA-based PCR clamping [11] (PNAClamp, Daejeon, Korea) before. For patients treated with osimertinib (Tagrisso, formerly AZD9291; AstraZeneca, Macclesfield, UK), samples from the latest biopsy before osimertinib treatment initiation were used. The research plan of the current study was approved by the Institutional Review Board (IRB) of Seoul at St. Mary’s Hospital (KC17DESI0147). At the time of sample collection, all patients provided a biobank-written informed consent form for the possible use of their samples in future research.

### 2.2. EGFR Mutation Tests

Tumor samples were collected and prepared as formalin-fixed, paraffin-embedded sides. Tumor DNA was extracted using a Maxwell 16 FFPE Tissue LEV DNA Purification Kit (Promega, Madison, WI, USA) for PANAMutyper tests (PANAGENE, Daejeon, Korea) and using a cobas DNA Sample Preparation Kit (Roche Molecular Systems, Pleasanton, CA, USA) for cobas v2 tests. Plasma was prepared by the Seoul St. Mary’s Hospital Biobank and stored at −70 °C. For this study, plasma samples were used to separate the supernatant for circulating free DNA (cfDNA) extraction. The cfDNA was extracted using QIAamp Circulating Nucleic Acid Kit (Qiagen, Hilden, Germany) and cobas cfDNA Sample Preparation Kit (Roche Molecular Systems, Pleasanton, CA, USA). Only DNA samples that passed qualitative and quantitative quality control (according to the kit manufacturers’ protocols) were used for *EGFR* mutation tests.

For PANAMutyper test, 5 μL DNA was added to 20 μL polymerase chain reaction (PCR) reagent (a mixture of 19 μL of peptide nucleic acid (PNA) probe and 1 μL of Taq DNA polymerase). PCR was carried out using the CFX96 Real-Time PCR Detection System (Bio-Rad, Hercules, CA, USA). The PCR-generated melting curves and the genotype of each sample were determined according to the specific fluorescence and melting temperature (Tm) of the melting curves. For cobas v2 test, DNA concentrations were set to 2 ng/μL and detected using a defined workflow channel in a cobas v2 4800 analyzer (Roche Molecular Systems, Pleasanton, CA, USA). Mutation analysis of cobas v2 was performed by Roche Korea using an algorithm specific to the cfDNA test. Both analyses were performed according to the manufacturers’ protocols.

### 2.3. Data Analyses

The concordance of the *EGFR* mutation tests was evaluated with cobas v2 as the reference using (1) positive percentage agreement (PPA) or sensitivity (the percentage of mutations in cobas v2 that is also detected in PANAMutyper); (2) negative percentage agreement (NPA) or specificity (the percentage of wild-type determined by cobas v2 that is also determined by PANAMutyper); (3) overall percentage of agreement (OPA) or accuracy (the percentage of wild-type plus mutations in cobas v2 that are also detected in PANAMutyper); and (4) Kappa coefficient (a statistical measure used to assess agreement between observers that provides more information than OPA because it takes into account chance agreement). A Kappa coefficient between 0.6 and 0.8 is generally regarded as “substantial agreement”, and a Kappa coefficient over 0.8 is generally regarded as “almost perfect agreement” [12]. Diagnostic precision in plasma was evaluated to indicate the percentage of mutations detected in plasma that were also detected in tumors. Tumor response was assessed by imaging techniques (such as computed tomography and magnetic resonance imaging) and determined based on the Response Evaluation Criteria in Solid Tumors (RECIST) 1.1 [13]. Survival time was defined as the period between osimertinib treatment initiation (or biopsy for patients who did not receive osimertinib) and the last follow-up (the study was closed on December 28 2019). All statistical analyses were performed with SPSS v.21 software (IBM SPSS, Armonk, NY, USA), and *p* value < 0.05 was considered statistically significant.

## 3. Results

### 3.1. Patients

A total of 244 eligible patients were identified (Figure 1). The ratio of male to female was 1.24:1, the median age was 66 years (range 29–85 years), and 133 were non-smokers (55%). Nearly half of the patients (51%) were in TNM stage III/IV (TNM: cancer staging system; T: primary tumor; N: lymph node; M: distant metastasis), and three-quarters of the patients had lung adenocarcinoma (76%). Nineteen tests from 17 patients were determined to be invalid (tests were invalid if samples failed to pass qualitative and quantitative quality control); 13 patients had invalid PANAMutyper test results (all from blood samples); 6 patients had invalid cobas v2 tests (5 from tumor samples and 1 from blood sample). These patients were excluded from data analyses. Considering the large sample size, both tests were considered successful in generating valid results.

Of the 227 patients with valid test results, the clinical characteristics of patients with *EGFR* wild-type were demographically different from those with *EGFR* activating mutations (Table 1). The median age of the wild-type group was five years older than that of the mutant group, and there were significantly less females, more smokers, and more squamous cell cancer in the wild-type group than the mutant group (*p* < 0.001 for all comparisons).

### 3.2. Concordance between Two Tests in Tumor and in Plasma

The concordant and discordant *EGFR* mutation test results in tumor tissue are shown in Table 2. In tumor tissue, the mutation detection rate was 43% by PANAMutyper and 37% by cobas v2. When cobas v2 was used as the reference for PANAMutyper, sensitivity (PPA) was 94%, specificity (NPA) was 90%, and accuracy (OPA) was 91%. Kappa coefficient was 0.82, indicating almost perfect agreement between tests in tumor samples.

In plasma, the mutation detection rate was 15% by PANAMutyper and 14% by cobas v2. For PANAMutyper, sensitivity (PPA) was 72%, NPA was 96%, and OPA was 93%. Kappa coefficient of the tests was 0.69, indicating a substantial agreement in plasma.

### 3.3. Concordance of Test Results between Tumor Tissue and Plasma

The *EGFR* mutation test results between tumor tissue and plasma were compared. For subjects with early-stage disease (stage I/II), only a small portion (0%–20%) of mutations found in tumor samples were also detected in plasma by PANAMutyper and cobas v2. Contrastingly, for subjects with advanced-stage disease (stage III/IV), test sensitivity (PPA) in plasma (with results in tumor samples as references) was markedly higher: around one-third of L858R and T790M and more than two-thirds of Ex19del in tumor can be detected in plasma (Figure 2A). For all disease stages, diagnostic precision for the three mutations in plasma (with results in tumor samples as references) was 100% by PANAMutyper and 86%–100% by cobas v2 (Figure 2B).

### 3.4. Re-Analyses of Discordant Results

Test results from 19 tumor samples were discordant between PANAMutyper and cobas v2 tests. These were compared to the original test results obtained using PNA clamping (Table 3). Ten tumor sample results were consistent between cobas v2 and PNA clamping, including nine wild-types and one Ex19del+T790M. Six tumor sample results were consistent between PANAMutyper and PNA clamping, including three L858R, one Ex19dels, one Ex19del+T790M, and one L858R+T790M. Two tumor sample results differed on all three tests. We attempted to verify the discordant results using dd-PCR. However, only six tumor samples had sufficient tissue for DNA extraction: four test results were in agreement with PANAMutyper, and two were in agreement with cobas v2.

### 3.5. Test Results in Patients with T790M Mutation

Of the 227 patients with valid test results, 47 patients were identified as T790M carriers by original PNA clamping test. These patients provided 46 tumor and 23 plasma samples (Figure 3). The concordance of PANAMutyper and cobas v2 results from this subgroup was evaluated using PPA only because all patients were diagnosed as *EGFR* mutants by the original PNA clamping test, and Kappa coefficient is inappropriate for skewed tests. The PPA (95% CI) of PANAMutyper was 91% (79%–97%) in tumor samples and 82% (52%–95%) in plasma. Among the 47 previously identified T790M carriers, 34 patients received osimertinib treatment. These 34 patients were at TNM stage IV and heavily treated at the time of osimertinib initiation (having previously received 3–10 lines of treatment). Of these patients, objective tumor response (complete response or partial response) was achieved in 74% (n = 25) of patients, and disease control was achieved in 88% (n = 30) of patients (Figure 3A). On the other hand, 16 patients were diagnosed with *EGFR* T790M mutation from the tumor samples by both PANAMutyper and cobas v2 but did not receive osimertinib treatment, including three patients who had early-stage disease without progression and 13 patients with late stage disease. Eight late-stage patients received tumor response evaluation, only one had stable disease, and the others showed progression. Six patients died, and seven patients were lost to follow-up (Figure 3B).

## 4. Discussion

This study is the first to compare PANAMutyper and cobas v2, two popular commercial *EGFR* mutation tests in Korea, for testing in both tumor and plasma samples from NSCLC patients. A large sample size of 244 patients was obtained, including 191 patients with paired samples (tumor and plasma). Results of the current study provide evidence that the two tests are equally accurate and can be used interchangeably.

A meta-analysis published in 2015 included over 30,000 NSCLC patients to create a “global *EGFR* mutMap” for global *EGFR* mutation frequency [14]. This study demonstrated that the overall *EGFR* mutation frequency is much higher in the Asia-Pacific area (47%) compared to Europe and America (15%–25%) and reported that the frequency in Korea is 43% (20%–56%), which is close to our result of 43% by PANAMutyper and 37% by cobas v2.

We observed significant demographic differences between those diagnosed with *EGFR* wild-type and those with *EGFR* mutations (Table 1). Consistent with previous reports [15,16,17], those with *EGFR* mutations were more likely to be female or non-smokers. In addition, patients without *EGFR* mutations were about five years older than those with *EGFR* mutations. This is probably because we included patients with squamous NSCLC who are less likely to have *EGFR* mutation but tend to be older than those with non-squamous NSCLC [18].

In this study, the two *EGFR* mutation tests were concordant in more than 90% of tumor and plasma samples (Table 2). The Kappa coefficient, a robust measurement for test agreement [12], showed that the two tests were in almost perfect agreement in tumors and in substantial agreement in plasma. Our results support use of the two tests interchangeably for *EGFR* activating mutation diagnosis in tumor samples.

The PNA-based PCR technology used by PANAMutyper has a lower limit of detection (>0.1%) [10] than the RT-PCR technology used by cobas v2 (>1%, according to cobas v2 label: https://www.accessdata.fda.gov/cdrh_docs/pdf12/P120019S007c.pdf). The difference in limit of detection between the two tests may explain why most of the inconsistent results (14 out of 19 samples) formed the same pattern: diagnosis of wild-type by cobas v2 but mutant by PANAMutyper (Table 3). Retesting these samples with a more sensitive method (such as dd-PCR) may validate the results. Unfortunately, only six of the discordant tumor samples had sufficient tissue remaining for dd-PCR retests, meaning that the sample size is too small to draw a conclusion for validation.

In the current study, the *EGFR* mutation detection rate in plasma was considerably low in early-stage cancer (stage I/II), similar to reports by other studies [19,20]. However, in late-stage cancer (stage III/IV), the two tests were able to detect over two-thirds of Ex19del and one-third of L858R and T790M mutations in tumors using plasma samples (Figure 2). Plasma detection rates by cobas v2 were consistent with a previous study [9].

Notably, despite a much lower detection rate in plasma than tumor, the diagnostic precision of both tests was very high (100% for PANAMutyper, 86%–100% for cobas v2). Therefore, although the current efficacy of osimertinib has only been established in patients with T790M-positive results in tumor, but not in patients with T790M-positive results only in plasma (according to cobas v2 label), the very high T790M diagnostic precision of both tests in plasma suggests that patients with T790M-positive plasma findings almost certainly have the mutation in tumor tissue also. Therefore, T790M-positive results in plasma should be considered sufficient evidence to initiate osimertinib treatment. Indeed, the latest guideline published by the International Association for the Study of Lung Cancer (IASLC) [21] and lung cancer education book published by the American Society of Clinical Oncology [4] place a higher priority on liquid biopsy over tumor biopsy in patients with progressive or recurrent disease.

In the subgroup of patients with previously diagnosed T790M mutation, PANAMutyper and cobas v2 tests were equally effective in identifying T790M mutation. Our real-world data in Figure 3 indicates that patients with T790M mutation respond favorably to osimertinib, with an impressive objective response rate of 74%. Contrastingly, those who did not receive osimertinib treatment showed very poor clinical prognosis and follow-up; for the most part, these patients were not treated by osimertinib because T790M mutations were diagnosed before osimertinib was approved or reimbursed through insurance in Korea (Figure 1). Although these osimertinib efficacy data are not from a randomized controlled study, they are consistent with a recently published phase III trial [3] and clearly show that osimertinib treatment, if guided by a reliable *EGFR* T790M mutation test, may lead to fundamentally superior clinical outcomes.

## 5. Conclusions

Two commercial *EGFR* mutation tests approved in Korea, cobas v2 and PANAMutyper, show highly concordant test results in both tumor and plasma samples from NSCLC patients. Although both tests show low sensitivity in plasma in early-stage disease, their high diagnostic precision in plasma make them attractive screening tools for identifying TKI treatment-feasible patients. Our study shows that around one-third or two-thirds (depending on the mutation) of the TKI treatment-feasible patients with late-stage NSCLC can be identified using liquid biopsy with both tests. Our real-world data reinforce the important role of reliable *EGFR* T790M mutation tests in guiding third-generation TKI treatments.

## Figures and Tables

**Figure 1 cancers-12-00785-f001:**
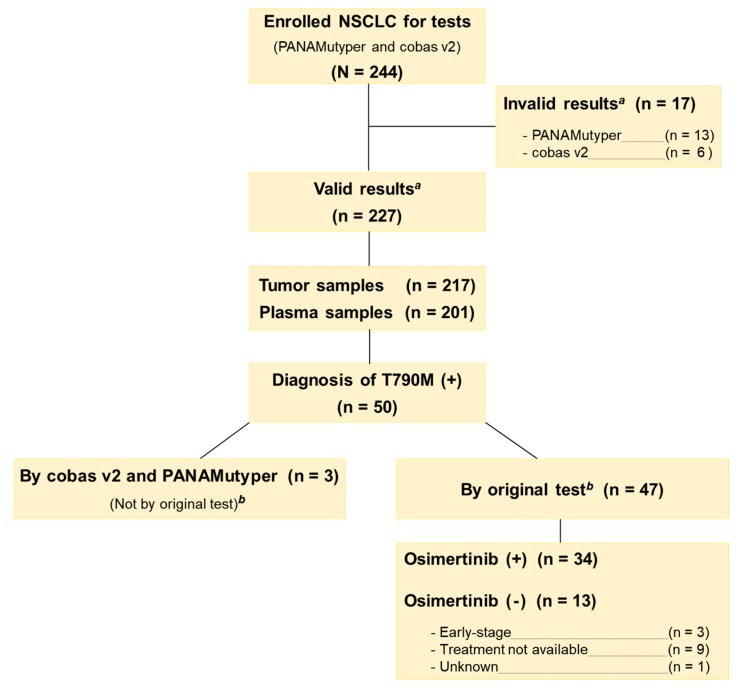
Consort diagram and study flowchart. ***^a^*** Tests were considered valid only if samples failed to pass qualitative and quantitative quality control. ***^b^*** Original test was peptic nucleic acid (PNA) clamping test.

**Figure 2 cancers-12-00785-f002:**
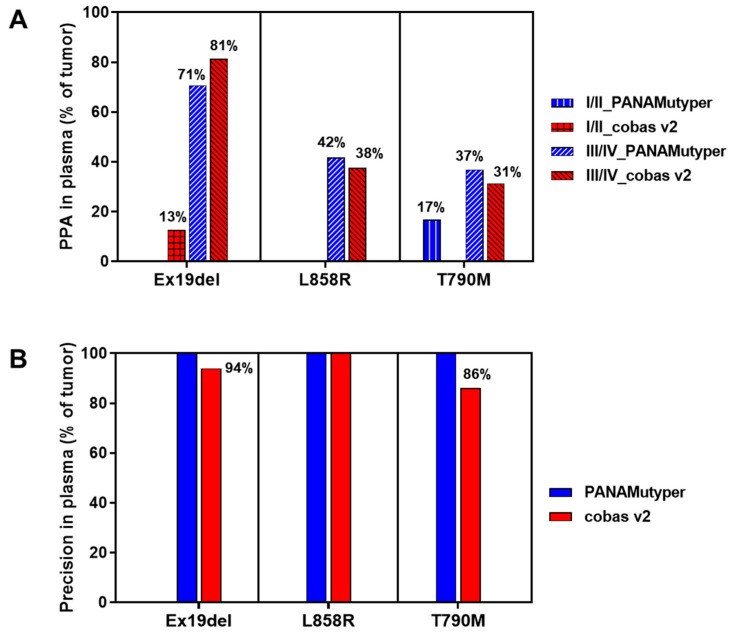
Detection of three *EGFR* mutations in plasma samples. (**A**) PPA of the two tests in plasma, stratified by disease stage. PPA is calculated as percentage of mutations detected in tumor that are also detected in plasma. (**B**) Precision of the two tests in plasma for all disease stages. Precision is calculated as the percentage of mutations detected in plasma that are also detected in tumor. PPA, positive percentage agreement; EGFR, epidermal growth factor receptor.

**Figure 3 cancers-12-00785-f003:**
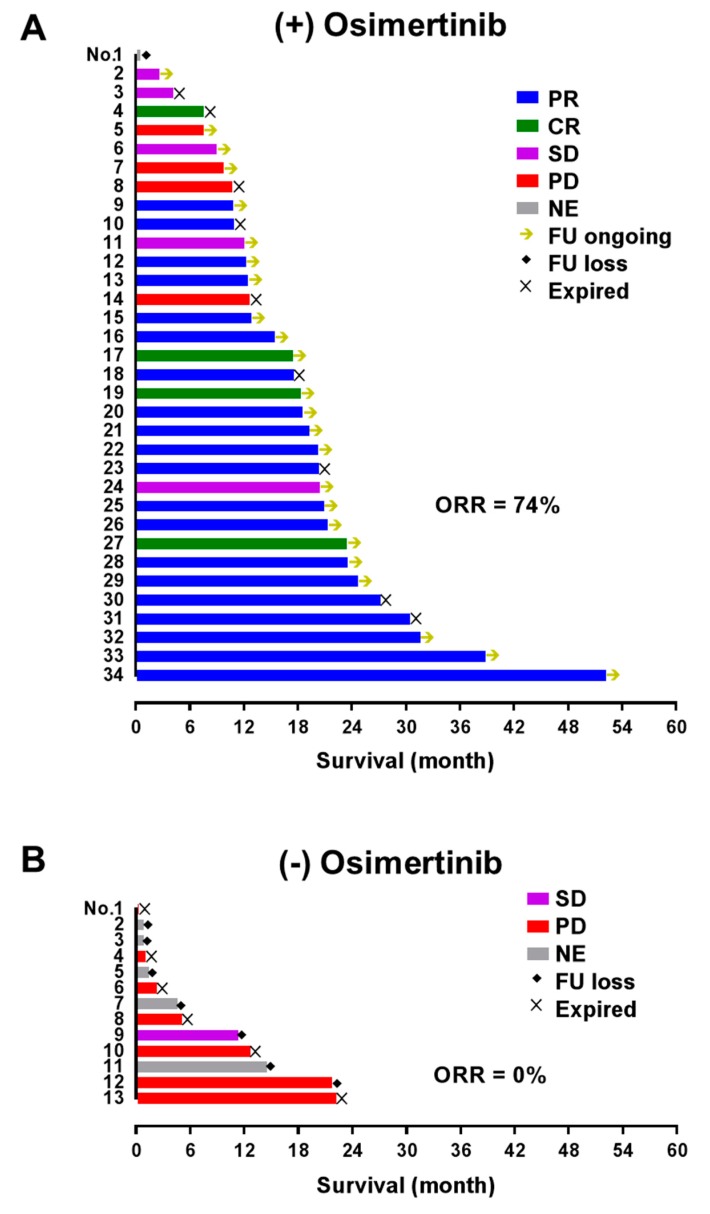
Survival and the best tumor response in patients with *EGFR* T790M mutation. (**A**) Patients who received osimertinib following development of drug resistance and T790M detection. All patients had previously received 3–10 lines of treatment, and osimertinib was used based on detection of T790M mutation using the PNA clamping test. (**B**) Patients with late-stage disease who were diagnosed with T790M mutation (by PNA clamping test: n = 10; by PANAMutyper and cobas v2: n = 3) but did not receive osimertinib. Arrows indicate ongoing follow-up; diamonds indicate loss to follow-up. PR, partial response; CR, complete response; SD, stable disease; PD, progressive disease; NE, not evaluated; FU loss, loss to follow-up; ORR, objective response rate.

**Table 1 cancers-12-00785-t001:** Characteristics of patients (n = 227) with valid *EGFR* test results using two mutation tests.

Patient Demographics (Wild vs. Mutant)	PANAMutyper Test Result			cobas v2 Test Result		
	Wild	Mutant	*p* value	Wild	Mutant	*p* Value
	(n = 130)	(n = 97)		(n = 143)	(n = 84)	
**Age *^a^***						
Median y (range)	68 (44–85)	63 (35–83)	*p* < 0.001	68 (44–85)	63 (35–83)	*p* < 0.001
**Sex *^b^***						
Male	89 (39%)	36 (16%)	*p* < 0.001	99 (44%)	26 (12%)	*p* < 0.001
Female	41 (18%)	61 (27%)		44 (19%)	58 (26%)	
**Smoking history *^b^***						
Never	57 (25%)	68 (30%)	*p* < 0.001	61 (27%)	64 (28%)	*p* < 0.001
Former	57 (25%)	24 (11%)		66 (29%)	15 (6.6%)	
Current	16 (7.0%)	5 (2.2%)		16 (7.0%)	5 (2.2%)	
**Histology *^b^***						
Adenocarcinoma	83 (37%)	91 (40%)	*p* < 0.001	90 (40%)	84 (37%)	*p* < 0.001
Squamous	45 (20%)	6 (2.6%)		51 (23%)	0	
Large cell	2 (0.9%)	0		2 (0.9%)	0	
**TNM stage *^b,c^***						
I	48 (21%)	33 (15%)	*p* = 0.644	53 (23%)	28 (12%)	*p* = 0.491
II	22 (9.7%)	9 (4.0%)		23 (10%)	8 (3.5%)	
III	29 (13%)	11 (4.8%)		34 (15%)	6 (2.6%)	
IV	31 (14%)	44 (19%)		33 (15%)	42 (19%)	

***^a^*** Independent *t*-test for *EGFR* wild vs. *EGFR* mutant groups diagnosed by the same assay. ***^b^*** Chi-square test for *EGFR* wild vs. *EGFR* mutant groups diagnosed by the same assay. ***^c^*** TNM stage was determined at the sample collection (TNM: cancer staging system; T: primary tumor; N: lymph node; M: distant metastasis).

**Table 2 cancers-12-00785-t002:** Concordance of two *EGFR* mutation tests in tumor tissue and plasma samples.

Test Results	Tumor Tissue	Plasma
(Reference: cobas v2)	(n = 217)	(n = 201)
**Concordant Results**		
**Both Wild**	123 (57%)	165 (82%)
**Both Mutant**	75 (35%)	21 (10%)
**E** **x19** **del**	17 (8.5%)	11 (5.5%)
**L858R**	14 (6.5%)	3 (1.4%)
**T790M**	1 (0.5%)	2 (1.0%)
**L858R+T790M**	30 (14%)	3 (1.5%)
**Ex19** **del+T790M**	13 (6.0%)	2 (1.0%)
**Discordant Result**		
**P_Mutant/c_Wild** ***^a^***	14 (6.5%)	7 (3.5%)
**c_Mutant/P_Wild** ***^b^***	1 (0.5%)	6 (3.0%)
**Different Mutant**	4 (1.8%)	2 (1.0%)
**Test Evaluation**		
**Sensitivity (PPA)** ***^c^*** **(95% CI)**	94% (86%–97%)	72% (54%–85%)
**Specificity (NPA)** ***^c^*** **(95% CI)**	90% (84%–94%)	96% (92%–98%)
**Accuracy (OPA)** ***^c^* (95% CI)**	91% (87%–94%)	93% (88%–95%)
**Kappa Coefficient** **(95% CI)**	0.82 (0.74–0.92)	0.69 (0.54–0.84)

***^a^*** P_Mutant/c_Wild: PANAMutyper result is mutant but cobas v2 result is wild; ***^b^*** c_Mutant/P_Wild: cobas v2 result is mutant but PANAMutyper result is wild; ***^c^*** Calculation with cobas v2 as reference; EGFR, epidermal growth factor receptor; OPA, overall percentage agreement; PPA, positive percentage agreement; NPA, negative percentage agreement.

**Table 3 cancers-12-00785-t003:** Reanalysis of discordance between PANAMutyper and cobas v2 tests.

No	TNM Stage	PANAMutyper	Cobas v2	PNA Clamping	dd-PCR
**1**	IIa	Ex19del	Wild	Ex19del	-
**2**	Ib	Ex19del	Wild	Wild	-
**3**	Ia	Ex20ins	Wild	Wild	-
**4**	IIIa	G719S	Wild	Wild	-
**5**	IIa	G719S	Wild	Wild	-
**6**	IIIb	G719S+T790M	Wild	Wild	-
**7**	IIIb	L858R	Wild	Wild	Wild
**8**	Ib	L858R	Wild	Wild	-
**9**	IIIb	L858R	Wild	Wild	L858R
**10**	Ia	L858R	Wild	L858R *^a^*	-
**11**	IV	L858R	Wild	L858R *^a^*	-
**12**	IV	L858R	Wild	L858R	L858R
**13**	IIIa	T790M	Wild	Ex20 Gln787Gln *^b^*	-
**14**	Ia	T790M	Wild	Wild	-
**15**	IV	L858R+T790M	T790M	L858R+T790M	L858R+T790M
**16**	Ia	G719C+S768I+T790M	G719X+S768I	-	-
**17**	IV	Ex19del	Ex19del+T790M	Ex19del+T790M	Ex19del+T790M
**18**	IV	Ex19del+T790M	Ex19del	Ex19del+T790M	Ex19del+T790M
**19**	IV	Wild	Ex19del	T790M	-

Underlined results indicate concordance with PANAMutyper, and framed results indicate concordance with cobas v2. ***^a^*** PNA clamping result is L858R or L861Q; ***^b^*** Synonymous mutation, c.2361G>A.

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
