# Peer review of "Evaluation of Two EGFR Mutation Tests on Tumor and Plasma from Patients with Non-Small Cell Lung Cancer"

_cancers, 2020, doi:10.3390/cancers12040785_

Round 1

Reviewer 1 Report

The study compares two commercial EGFR mutations testing applications available in South Korea. The authors showed similarity in the diagnostic capacity of both tests with concordant results in sensitivity, specificity and accuracy. Importantly, the data also suggested the utility of plasma based testing for T790M alterations

Major revision

1. Table 1 is very confusing. What do the numbers add up to? The way the table is presented, despite a sample size of 227, it appears that there were 273 patients with WT results and 181 patients with MT EGFR. I would recommend Switching the headings for the tables to PANAMutyper and Cobasv2, with subheadings for each as WT and MT.

Author Response

Reply

Thank you very much for your constructive feedback. Based on your suggestion, we revised Table 1 as follow:

Table 1. Characteristics of patients (n=227) with valid EGFR test results using two mutation tests.

PANAMutyper Test Result

cobas v2 Test Result

Wild

(n=130)

Mutant

(n=97)

P value

Wild

 (n=143)

Mutant

 (n=84)

P value

Agea

Median yrs (range)

68 (44-85)

63 (35-83)

P<0.001

68 (44-85)

63 (35-83)

P<0.001

Sexb

Male

89 (39%)

36 (16%)

P<0.001

99 (44%)

26 (12%)

P<0.001

Female

41 (18%)

61 (27%)

44 (19%)

58 (26%)

Smoking historyb

Never

57 (25%)

68 (30%)

P<0.001

61 (27%)

64 (28%)

P<0.001

Former

57 (25%)

24 (11%)

66 (29%)

15 (6.6%)

Current

16 (7.0%)

 5 (2.2%)

16 (7.0%)

 5 (2.2%)

Histologyb

Adenocarcinoma

83 (37%)

91 (40%)

P<0.001

90 (40%)

84 (37%)

P<0.001

Squamous

45 (20%)

 6 (2.6%)

51 (23%)

0

Large cell

 2 (0.9%)

0

 2 (0.9%)

0

TNM stageb,c

I

48 (21%)

33 (15%)

P=0.644

53 (23%)

28 (12%)

P=0.491

II

22 (9.7%)

 9 (4.0%)

23 (10%)

 8 (3.5%)

III

29 (13%)

11 (4.8%)

34 (15%)

 6 (2.6%)

IV

31 (14%)

44 (19%)

33 (15%)

42 (19%)

aIndependent t-test for EGFR wild vs. EGFR mutant groups diagnosed by the same assay. bChi-square test for EGFR wild vs. EGFR mutant groups diagnosed by the same assay. cTNM stage was determined at the sample collection.

In addition, we also made some minor changes (such as font styles, figure legends, information of drug and equipment) which are shown in red. Thank you for helping us improve this manuscript.

Reviewer 2 Report

This manuscript describes the evaluation of two EGFR mutation tests, cobas v2 and PANAMutyper, for the detection of EGFR activating mutations Ex19del, L858R, and T790M in tumor tissue and plasma derived from 244 non-small cell lung cancer (NSCLC) patients. The authors concluded that “the two tests were in almost perfect agreement in tumors” and “our results support the use of the two tests interchangeably for EGFR activating mutation diagnosis in tumor samples.” Overall, the manuscript is convincing in its comprehensive methodical approach. Although a couple of issue need to be clarified, there is sufficient information currently in this manuscript to warrant publication.

Comments:

  • Please add information about the administration of first-line osimertinib therapy to NSCLC patients with EGFR activating mutations. The emergence of the T790M mutation should not be detected in these cases, and the meaning of T790M detection might have changed.
  • I agree that EGFR mutation testing in plasma is an attractive screening tool, but its clinical application might be challenging based on the authors’ results.

Author Response

Reply:

  1. Thank you very much for your comment. In our study, none of the 34 patients took osimertinib as first-line treatment, and all of them had previously received 3-10 lines of treatment. However, all these patients started the drug based on the emergence of T790M mutation. In order to present this information more clearly, we have added more description into the legend of Figure 3 (attached below).

  1. We appreciate your comment. Although liquid biopsy shows low sensitivity in patients with early stage NSCLC, in patients with late stage disease, liquid biopsy is able to detect 1/3 to 2/3 EGFR activating mutations (Ex19del, L858R, or T790M) that are detected in tumors. More importantly, based on our results, once a patient is diagnosed with a mutation using liquid biopsy, the patient can avoid having an additional tumor biopsy because the liquid biopsy shows very high precision (86%-100%). Therefore, we expect that 1/3 to 2/3 patients with late stage disease can benefit from the liquid biopsy.

In addition to your comments, we also made some minor changes (such as font styles, figure legend, information of drug and equipment) which are shown in red. Thank you for helping us improve this manuscript.

Figure 3. Survival and the best tumor response in patients with EGFR T790M mutation. A, Patients who received osimertinib following development of drug resistance and tumor biopsy. All patients had previously received 3-10 lines of treatment, and osimertinib was used based on detection of T790M mutation using the PNA clamping test. B, Patients who did not receive osimertinib following tumor biopsy. Arrows indicate ongoing follow-up, diamonds indicate loss to follow-up. PR, Partial response; CR, complete response; SD, stable disease; PD, progressive disease; NE, not evaluated; FU loss, loss to follow-up; ORR, objective response rate.
